# Coupled MOP and PLUS-SA Model Research on Land Use Scenario Simulations in Zhengzhou Metropolitan Area, Central China

Pengfei Guo [1,2,3], Haiying Wang [1,2,4,*], Fen Qin [1,2,4], Changhong Miao [2,5] and Fangfang Zhang [6]

[1] College of Geography and Environmental Science, Henan University, Kaifeng 475004, China; 104753151243@vip.henu.edu.cn (P.G.); qinfen@henu.edu.cn (F.Q.)

[2] Key Laboratory of Geospatial Technology for the Middle and Lower Yellow River Regions, Henan University, Ministry of Education, Kaifeng 475004, China; chhmiao@henu.edu.cn

[3] Institute of Geographic Sciences and Natural Resources Research, CAS, Beijing 100101, China

[4] Institute of Urban Big Data, Henan University, Kaifeng 475004, China

[5] Key Research Institute of Yellow River Civilization and Sustainable Development, Henan University, Kaifeng 475004, China

[6] School of Resources and Environment/Collaborative Innovation Center of Urban-Rural Coordinated Development, Henan University of Economics and Law, Zhengzhou 450046, China; zhangff.19b@igsnrr.ac.cn

* Correspondence: whyhdgis@henu.edu.cn

**Abstract:** Land use simulations are critical in predicting the impact of land use change (LUC) on the Earth. Various assumptions and policies influence land use structure and are a key factor in decisions made by policymakers. Meanwhile, the spatial autocorrelation effect between land use types has rarely been considered in existing land use spatial simulation models, and the simulation accuracy needs to be further improved. Thus, in this study, the driving mechanisms of LUC are analyzed. The quantity demand and spatial distribution of land use are predicted under natural development (ND), economic development (ED), ecological protection (EP), and sustainability development (SD) scenarios in Zhengzhou based on the coupled Multi-Objective Programming (MOP) model and the Patch-generating Land Use Simulation model (PLUS) considering Spatial Autocorrelation (PLUS-SA). We conclude the following. (1) The land use type in Zhengzhou was mainly cultivated land, and 83.85% of the land for urban expansion was cultivated land from 2000 to 2020. The reduction in forest from 2010 to 2020 was less than that from 2000 to 2010 due to the implementation of the policy in which farmland is transformed back into forests. (2) The accuracy of PLUS-SA was better than that of the traditional PLUS and Future Land Use Simulation (FLUS) models, and its Kappa coefficient, overall accuracy, and FOM were 0.91, 0.95, and 0.29, respectively. (3) Natural factors (temperature, precipitation, and DEM) contributed significantly to the expansion of cultivated land, and the increase in forest, grass, and construction land was greatly affected by socioeconomic factors (population, GDP, and proximity to town). (4) The land use structure will be more in line with the current requirements for sustainable urban development in the SD scenario, and the economic and ecological benefits will increase by $0.75 \times 10^4$ billion CNY and 1.71 billion CNY, respectively, in 2035 compared with those in 2020. The PLUS-SA model we proposed had higher simulation accuracy in Zhengzhou Compared with the traditional PLUS and FLUS models, and our research framework can provide a basis for decision-makers to formulate sustainable land use development policies to achieve high-quality and sustainable urban development.

**Keywords:** land use change; urban expansion; PLUS-SA model; scenario simulation; Zhengzhou city

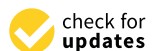



## 1. Introduction

Land is a necessity for people's survival and an indispensable factor of production for social development [1,2]. As a critical climate variable, land use change (LUC) is the result of the natural environment, human activities, and policy [3,4] and has been widely studied

in Earth system modeling, natural resource management, and ecological conservation. It has a crucial impact on global climate change and sustainable development [5–7]. In recent years, with an increase in the world's population and an acceleration in urbanization, the patterns and types of land use have changed significantly, leading people to pursue economic development while also facing serious challenges in the form of regional or global ecological and environmental issues [8], such as land degradation and desertification [9], a sharp reduction in cropland and deforestation [10], and a rapid loss of biodiversity [11]. The pressure on land resources has become increasingly prominent. Located in the hinterland of the Central Plains, Zhengzhou City is the main city of the Central Plains Urban Agglomeration, has superior natural conditions, and benefits from many economic advantages [12,13]. However, while land use changes caused by population growth, urban expansion, and rapid urbanization in Zhengzhou City have promoted economic growth, the ecological environment has been subject to an increasing level of threat [14–16]. Thus, the subject of how to utilize limited land resources to promote the coordinated development of the social economy and ecological protection, as well as maximize the comprehensive benefits and sustainable development of regional land use, has become a key concern in the field of land use research and in relevant government departments [17–19].

Land use simulation and prediction are important elements of land science and important ways to scientifically combine various types of land resources and optimize the spatial layout under certain constraints [20,21] while also being a fundamental guarantee for the sustainable development of land resources [22,23]. For a long time, scholars in the field of land use research have developed various LUC modeling methods according to the research scale of LUC, disciplinary background, and the key research field of the research team [24–26]. Each model has potential and boundedness for the needs of land planners and decision-makers. Research on LUC simulation at home and abroad has experienced a development process from quantitative models to spatial simulations, from simulations of one type of land use to dynamic changes in multiple land uses, and from a single model to a coupled model [27]. A quantity prediction model explains how the quantity of different land types changes; in other words, solving the "How" problem. Representative models include logistic regression [28], neural network (ANN) [29], Markov [30], system dynamics (SD) [31], multi-objective linear programming (MOP) [32], and so on. A spatial model focuses more on explaining the spatial variation in different land types; that is, solving the "Where" problem using typical multi-agent models (MAS) [33] and models that have been developed based on the cellular automata (CA) theory [34]. A single quantitative or spatial model cannot simulate the complex process of LUC more accurately; for example, a quantitative model depicts the structural requirements of land use from a macro perspective, but lacks the ability to process spatial factors and the feedback of each element in space [35]. Conversely, spatial models determine the evolution of a system from a local perspective. They cannot properly reflect the macro-demand of different land use types and the impacts of government policies and climate background [36]. Therefore, coupling the "top-down" quantitative model and the "bottom-up" spatial model results in the current mainstream model of land use simulation [37].

CA is extensively applied in LUC spatial simulation because of its powerful spatiotemporal coupling characteristics and spatial computing functions, which can be used to explore complex geographic processes [38–40]. Models developed based on CA, for instance, CA-Markov [41], Conversion of Land Use and its Effects (CLUE) [42], Conversion of Land Use and its Effects at Small regional extent (CLUE-S) [43], SLEUTH (Slope, Land use, Excluded, Urban, Transportation, Hill shade) [44], and FOREcasting SCEnarios of Land-use Change (FORE-SCE) [25], are typical. However, although they can be used to visualize the spatial model of LUC, they do not provide an optimal solution [24]. Thus, the Future Land Use Simulation (FLUS) model constructed by Liu et al. [24] adopted a roulette wheel selection mechanism and an adaptive inertia mechanism in the CA model to dispose of the complex competition and interaction between different land use types. Nevertheless, Liang et al. [19] pointed out that the current CA-based models still have

the following shortcomings: (1) existing CA models are still insufficient with regard to revealing the intrinsic driving force of LUC and (2) simulating multiple land use types dynamically is difficult with space and time patch-level changes. Based on this, Liang et al. [19] developed a patch-generating land use simulation (PLUS) model and used this model for LUC simulation in Wuhan. The number of studies based on the PLUS model has gradually increased over time [45–47]. Numerous studies have shown that land use simulations based on the PLUS model can obtain higher simulation accuracy and closely resemble real landscape pattern indicators [19,46,48]. This provides a base for land use planners to formulate sustainable land management policies [49–51].

Although much has been achieved regarding land use simulations at home and abroad, existing research rarely considers the possible spatial autocorrelation effect in land use data [52]. Some scholars have proposed an auto logistic model by introducing the spatial autocorrelation factor into a logistic regression model to capture the influence of the spatial autocorrelation effect on LUC, thereby improving the model simulation accuracy [53,54]. Therefore, it is urgent to introduce spatial autocorrelation factors to the LUC simulation process to obtain more accurate simulation results. Additionally, existing land use simulation studies usually focus on improving the simulation accuracy by optimizing algorithms, but lack any understanding of the LUC variation mechanism [19]. MOP breaks through the only limitation of the objective function and can optimize the allocation of land use structure under the premise of taking into account economic and ecological benefits according to different policies in order to maximize regional comprehensive benefits [55–57]. The PLUS model identifies the rules of LUC and mines their drivers while also having a higher degree of simulation accuracy compared with other CA models [48,50]. Thus, with the goal of coordinating economic development and ecological protection, this paper establishes natural development (ND), economic development (ED), ecological protection (EP), and sustainability development (SD) scenarios for simulating future multi-scenario land use patterns by combining coupled MOP and PLUS-SA models (see Section 2.3.3 for details) with regional development goals and realization paths. This research will be able to seek a future sustainable land structure and spatial layout scenario, and then provide a reference for promoting the optimal allocation of regional land resources.

## 2. Materials and Methods

### 2.1. Study Area

Zhengzhou City (112°42′E–114°14′E, 34°16′N–34°58′N)—the political, economic, and cultural center of Henan Province, a national central city, and a crucial transportation and communication hub in China—is located in the north–central part of Henan Province, at the boundary between the middle and lower reaches of the Yellow River. Meanwhile, Zhengzhou City, with a large population and fast economic development, is the core city of the Central Plains City group. People ignore the protection of the regional ecological environment while pursuing the economic benefits of land. In recent years, due to the influence of human activities, the LUC in Zhengzhou has become more significant. Therefore, this study takes the metropolitan area of Zhengzhou as the research area to simulate the future land use layout of Zhengzhou under different scenarios in order to provide references for the sustainable development of land resources in Zhengzhou. The general topographic trend of Zhengzhou is high in the southwest and low in the northeast, descending in steps (Figure 1). The climate in Zhengzhou can be categorized as a Monsoon Climate of Medium Latitudes, with an annual average temperature of 14.7 °C and an average precipitation of 632.4 mm. At the end of 2022, the permanent population of Zhengzhou reached 12.828 million. Zhengzhou's Gross Domestic Product (GDP) was 1293.4 billion Chinese Yuan (CNY) at constant prices, an increase of 1.0% over the previous year. The added value of primary industry was 18.56 billion CNY, up by 3.7%; the added value of secondary industry was 517.46 billion CNY, up by 2.0%; the added value of tertiary industry was 757.45 billion CNY, up by 0.2%. The tertiary industrial structure was 1.4:40.0:58.6. As of December 2022, Zhengzhou had jurisdiction over six districts, five county-level cities, and one county, with

a total area of approximately 7567 km$^2$ and an urbanization rate of about 78.4%. In recent years, with the development of the social economy and improvements in the urbanization level, the LUC in Zhengzhou has become more obvious, especially with regard to the rapid expansion of construction land. The urban built-up area of Zhengzhou increased from 744.77 km$^2$ in 2016 to 1284.89 km$^2$ in 2021, an increase of 540.12 km$^2$ in five years.

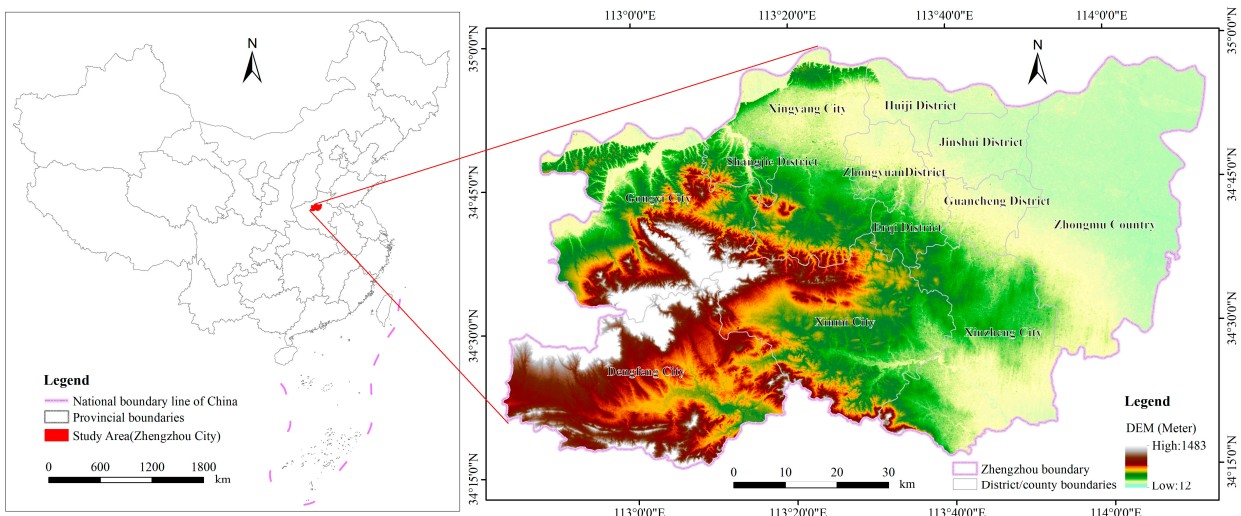

**Figure 1.** The geographical location of Zhengzhou.

### 2.2. Data Sources

The 30 m resolution land use data sets of Zhengzhou in 2000, 2010, and 2020 were derived from the Chinese Academy of Sciences, Resources, and Environment Science Data Center (https://www.resdc.cn/, accessed on 12 June 2023). The data set was mainly based on Landsat satellite remote sensing data and constructed by human–machine interactive visual interpretation. Landsat-TM/ETM remote sensing image data were mainly used for the interpretation of remote sensing data from 2000 and 2010, while Landsat8 OLI remote sensing image data were the data source for 2020. Landsat8 is the latest satellite in the U.S. government's Landsat family, which launched on 11 February 2013, carrying the OLI Land Imager and TIRS thermal infrared sensor. Landsat8's OLI Land Imager consists of nine bands. OLI includes all bands of the ETM+ sensor, in addition to two new bands, namely the blue band (band1; 0.433–0.453 μm), which is mainly used for coastal zone observation, and the short-wave infrared band (band9; 1.360–1.390 μm), which includes strong water vapor absorption features and can be used to detect clouds. In the process of land use remote sensing monitoring in 2020, standard false-color composite images synthesized by OLI land imager in 5, 4, and 3 bands were used for artificial visual interpretation. According to the natural properties of land resources, the data set can be divided into six first-level categories: cultivated land, forestland, grassland, water bodies, construction land, and unused land (Figure 2), and the accuracy of all of them is above 90%. This data set is recognized as a land use data set with high precision, high resolution, and a long time series.

The socioeconomic data were obtained from the China Agricultural Product Price Survey Yearbook and Henan Statistical. Some government reports and planning drawings were taken from the Zhengzhou Municipal People's Government website. The geographic and socioeconomic data used in the study concerning elements such as elevation, temperature, precipitation, soil type, population, GDP, points of interest (POI), and traffic data were obtained from different websites. For details concerning the year, resolution, and source of all data, see Table 1.

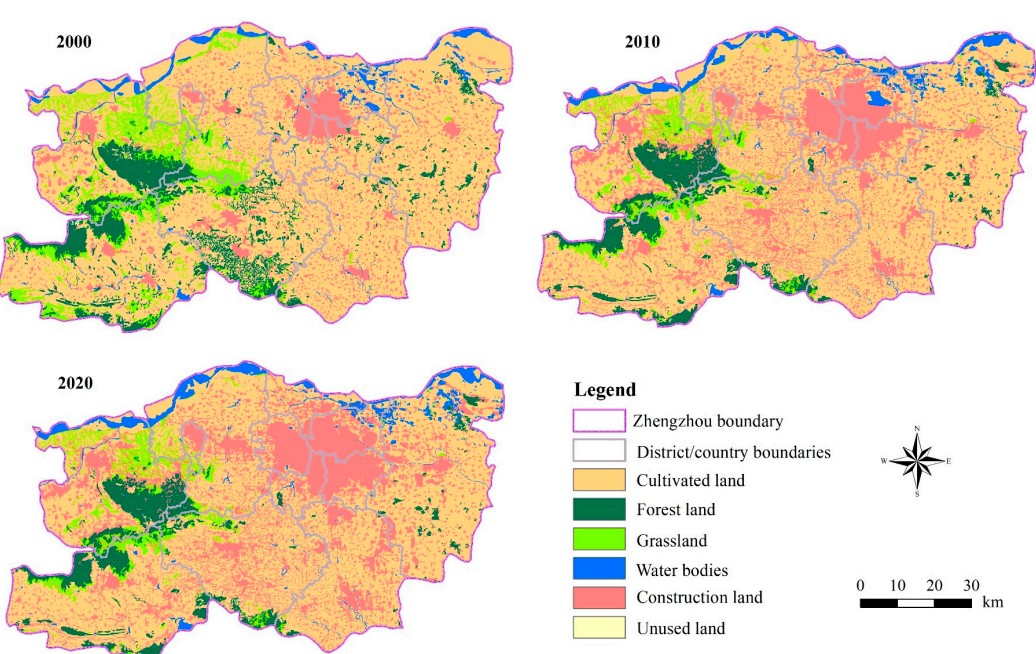

**Figure 2.** Land use data set (2000, 2010, and 2020) for Zhengzhou.

**Table 1.** Details of the data used in this study.

| Data Types | Data Name | Years | Attributes/Resolution | Data Sources |
|---|---|---|---|---|
| Government report | integrated land-use planning | 2006–2020 | Text, atlas | Zhengzhou Municipal People's Government network (http://www.zhengzhou.gov.cn/, accessed on 12 June 2023) |
| Statistic data | Socioeconomic data | 2000–2020 | Excel | Henan Provincial Bureau of Statistics (http://www.ha.stats.gov.cn/, accessed on 12 June 2023) |
| Land use | Land use data | 2000–2020 | TIFF/30 m | Resource and Environmental Sciences and Data Center (http://www.resdc.cn/, accessed on 12 June 2023) |
| Socioeconomic data | Traffic data | 2020 | SHP | Open Street Map (https://www.openstreetmap.org/, accessed on 12 June 2023) |
| | Population | 2015 | TIFF/1 km | Global Change Scientific Research Data Publishing & Repository (http://www.resdc.cn/, accessed on 12 June 2023) |
| | GDP | 2015 | TIFF/1 km | |
| | POI | 2020 | SHP | Baidu Map crawler (https://map.baidu.com/, accessed on 12 June 2023) |
| Climate data | Temperature | 2018 | TIFF/1 km | National Meteorological Science Data Center (http://data.cma.cn/, accessed on 12 June 2023) |
| | Average annual precipitation | 2018 | TIFF/1 km | |
| Natural conditions | DEM | 2015 | TIFF/30 m | NASA SRTM1 v3.0 |
| | Slope, aspect | 2015 | TIFF/30 m | Extracted from DEM data |
| | Aspect | 2015 | TIFF/30 m | |
| | Soil type | 2012 | SHP | National Earth System Science Data Center (http://www.geodata.cn/, accessed on 12 June 2023) |

*2.3. Research Methods*

Four ND, ED, EP, and SD scenarios were established. Among them, the ND scenario follows the natural evolution law of land use structure from 2000 to 2020. The ED scenario takes the improvement of regional economic benefits as its ultimate goal, focuses on developing land use types with high economic benefits, strengthens urban infrastructure

construction, and promotes a steady increase in the rate of urbanization. The EP scenario strengthens the protection of nature reserves, forestland, water bodies, and other land with important ecological functions, ensures biodiversity and environmental quality and safety, and takes the increase in ecological benefits as the ultimate goal. The SD scenario focuses on the coordinated development of economic construction and ecological protection. It promotes high-quality economic development on the precondition of the high-level protection of the ecological environment to maximize ecological and economic benefits.

Figure 3 shows the coupled model framework, which contains the procedures of the MOP and PLUS-SA models. First, we used the Markov model to predict the land use structure demand in Zhengzhou City in 2035 under the ND scenario. Based on the MOP model, Lingo 12.0 software was used to obtain the land use structure demand in ED, EP, and SD scenarios. Then, natural conditions, traffic locations, the social economy, and other driving factors were introduced in order to simulate the land use patterns under the four scenarios of Zhengzhou in 2035 by utilizing the FLUS, PLUS, and PLUS-SA models, respectively. This study can verify that our PLUS-SA model possesses a higher degree of simulation accuracy than the FLUS and traditional PLUS models. It can also function in line with the current requirements for high-quality and sustainable urban development land use structure and spatial distribution scenarios. This will assist relevant government departments in formulating sustainable land policies.

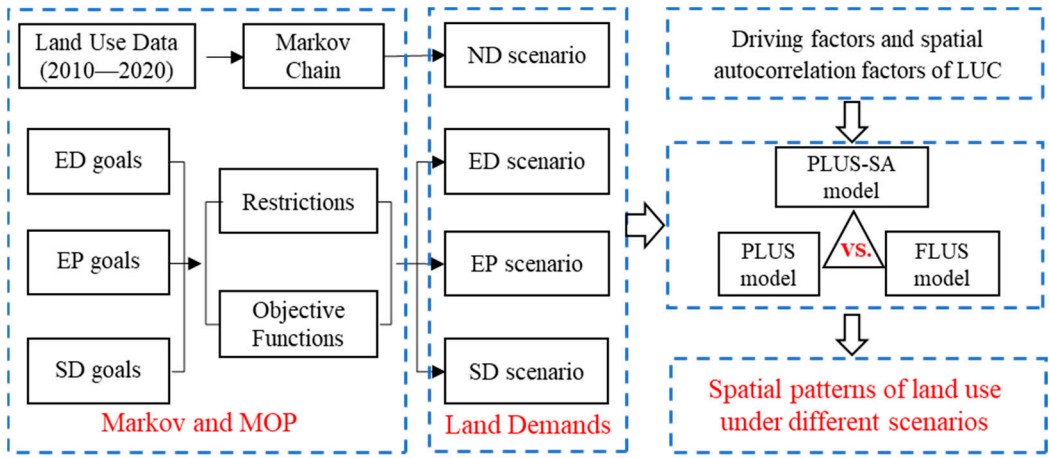

**Figure 3.** Coupled model framework including the MOP and PLUS-SA models.

### 2.3.1. Research Methods of LUC

Single land use dynamics ($K$) can intuitively reflect the change amplitude and velocity of a single land type during a research period, and the comprehensive land use dynamic degree ($LC$) can reflect the overall change in the land use quantity of a study region within a certain time range. Therefore, the single and comprehensive land use dynamic degrees were adopted in order to explore the LUC in Zhengzhou from 2000 to 2020. The comprehensive land use dynamic degree in Zhengzhou was visualized from a spatial perspective by creating a $2 \times 2$ km fishing net.

$$K_i = \frac{M_a - M_b}{M_a} \times \frac{1}{T_2 - T_1} \times 100\% \qquad (1)$$

$$LC = \left[ \frac{\sum_{i=1}^{n} \Delta LU_{i-j}}{2 \sum_{i=1}^{n} LU_i} \right] \times \frac{1}{T} \times 100\% \qquad (2)$$

where $K_i$ represents the single dynamic degree of land type '$i$' from the start date $T_1$ to the end date $T_2$. $M_a$ and $M_b$ represent the area of land use type '$i$' on $T_1$ and $T_2$, respectively. $T_2 - T_1$ represents the length of the monitoring period. Thus, when $K_i$ is greater than 0, the area of land type '$i$' increases within the monitoring period; otherwise, it decreases.

Therefore, the larger the absolute value of $K_i$, the greater the increase or decrease, and the faster the rate. In Formula (2), $LC$ represents the comprehensive land use dynamic degree, $LU_i$ is the area of land type '$i$' on T$_1$, and $\Delta LU_{i-j}$ is the area of land type '$i$' converted to '$j$'. In this study, the $LC$ in Zhengzhou City was visualized by using "Create Fishnet" in ArcGIS.

### 2.3.2. Multi-Objective Programming Model

The Multi-Objective Programming (MOP) model is a mathematical model for researching geography and regional development, and it aims to derive optimal solutions by considering multiple objective functions simultaneously [17,58]. The MOP model can predict the land use structure more scientifically and reasonably based on objective laws and constraint data, and has been widely used in land use simulation research [59]. The MOP model includes decision variables, objective functions, and constraint conditions [18]. It focuses on the decision to make one or more goals achieve the best value under subjective or objective conditions.

$$f_1(x) = max \sum_{j=1}^{n} Eco_j \times x_j \tag{3}$$

$$f_2(x) = max \sum_{j=1}^{n} Esv_j \times x_j \tag{4}$$

$$f_3(x) = max\{f_1(x), f_2(x)\} \tag{5}$$

$$s.t. = \begin{cases} \sum_{j=1}^{n} a_{ij}x_j = (\geq, \leq)b_j, (i = 1, 2, \dots, m) \\ x_j \geq 0, (j = 1, 2, \dots, n) \end{cases} \tag{6}$$

where $f_1(x)$, $f_2(x)$, and $f_3(x)$ represent the objective functions under the ED scenario, EP scenario, and SD scenario, respectively; and $x_j$ is the area of the $j^{\text{th}}$ decision variable. There are six decision variables in this study: $x_1$ cultivated land, $x_2$ forestland, $x_3$ grassland, $x_4$ water bodies, $x_5$ construction land, and $x_6$ unused land. $Eco_j$ and $Esv_j$ are the economic and ecological benefit coefficients of different land uses per unit area, respectively. Under the constraint condition $s.t.$, $a_{ij}$ is the coefficient corresponding to the $j^{\text{th}}$ variable under the $i^{\text{th}}$ constraint condition, and $b_j$ is the constraint value.

(1) Determination of the objective function: According to the output per unit area of each land type in Zhengzhou from 2000 to 2020, the economic benefit coefficient of each land type was calculated (Table 2). Among them, the total output values of agriculture, forestry, animal husbandry, and fisheries were used to estimate the economic benefits of cultivated land, forestland, grassland, and water bodies, respectively. The GDPs of the secondary and tertiary industries were used to estimate the economic benefits of construction land and the economic benefits of unused land were assigned a value of 0 [19,55]. Therefore, Formula (3) can be expressed as:

$$f_1(x) = max(6.54x_1 + 1.86x_2 + 189.72x_3 + 1.68x_4 + 1890.36x_5 + 0x_6) \tag{7}$$

**Table 2.** Economic and ecological benefit coefficients per unit area ($10^6$ CNY/km$^2$).

| Efficiency Coefficient | Cultivated Land | Forestland | Grassland | Water Bodies | Construction Land | Unused Land |
|---|---|---|---|---|---|---|
| $Eco_j$ | 6.54 | 1.86 | 189.72 | 1.68 | 1890.36 | 0 |
| $Esv_j$ | 1.19 | 4.23 | 1.76 | 7.62 | 0 | 0.21 |

The ecological benefits of each land type were estimated by utilizing the Xie et al. [60, 61] "Equivalent value of ecological services per unit area of ecosystems in China" table.

The calculation of the economic value of food production services per unit area of the farmland ecosystem can be seen in Guo et al. [6]. Meanwhile, the cultivated land, forestland, and grassland were revised grid-by-grid using NDVI (the NDVI values of construction land, water bodies, and unused land were almost 0, which were not considered here). Finally, the ecological benefit coefficients of six land types were acquired (Table 3). Therefore, Formula (4) can be expressed as:

$$f_2(x) = \max(1.19x_1 + 4.23x_2 + 1.76x_3 + 7.62x_4 + 0x_5 + 0.21x_6) \tag{8}$$

**Table 3.** The constraint conditions of various land types in Zhengzhou.

| Constraint Class | Constraint Conditions | Evidence and Description |
|---|---|---|
| Total area constraints | $\sum_{j=1}^{6} x_j = A = 7567.80 \text{ km}^2$ | The sum of the six land use types area must equal the total area of Zhengzhou City. |
| Total population constraint | $75.81 (x_1 + x_2 + x_3) + 6744.06$ $x_5 \leq 1.8 \times 10^7$ | By 2035, the population of Zhengzhou is expected to be 18 million. According to Wang et al. [55], the population densities on agricultural land (cropland, woodland, and grassland) and construction land will be 75.81 and 6744.06 people per square kilometer by 2035. |
| Cultivated land area constraint | $0.4901 \leq x_1/A \leq 0.5632$ | With improvements in farming technology, the yield of cultivated land per unit area increases, which can effectively compensate for the food security problems caused by the expansion of construction land to cultivated land. Therefore, 56.32% (the proportion of cultivated land in 2020) is set as the upper limit and 49.01% (the predicted proportion of cultivated land in 2035 from the Markov chain) is set as the lower limit for the percentage of cultivated land in 2035. |
| Forestland area constraint | $0.0741 \leq x_2/A \leq 0.0889$ | Forestland should account for 7.41% (the proportion of forestland in 2020) to 8.89% (1.2 times the forestland area in 2020) of the total area. |
| Grassland area constraint | $0.0499 \leq x_3/A \leq 0.0527$ | 5.27% (cover degree of grassland in 2010) is set as the upper limit and 4.99% (predicted cover degree of grassland in 2035 from the Markov chain) is set as the lower limit for the percentage of grassland in 2035. |
| Water bodies area constraint | $0.0372 \leq x_4/A \leq 0.0496$ | In order to protect water resources with high ecosystem service value coefficients, the area of water bodies in Zhengzhou is required to be no less than 90% and no greater than 120% of the 2020 level. |
| Construction land area constraint | $0.2766 \leq x_5/A \leq 0.4150$ | The percentage of construction land will be between 80% and 120% of the predicted construction land in 2035 from the Markov chain. |
| Non-negative constraint | $x_j \geq 0, j = 1, 2, \ldots 6$ | The area of each land type is non-negative. |

After measuring the economic benefits $f_1(x)$ and ecological benefits $f_2(x)$ of land resources in Zhengzhou, the objective function $f_3(x)$ under the SD scenario can be expressed as:

$$f_3(x) = max\{(6.54x_1 + 1.86x_2 + 189.72x_3 + 1.68x_4 + 1890.36x_5 + 0x_6)\varphi_1 \\ + (1.19x_1 + 4.23x_2 + 1.76x_3 + 7.62x_4 + 0x_5 + 0.21x_1)\varphi_2\} \tag{9}$$

In Formulas (7)–(9), $x_1$–$x_6$ denote cultivated land, forestland, grassland, water body, construction land, and unused land, respectively. The future development orientation of Zhengzhou City and the goal of simultaneously improving economic and ecological benefits were taken into account. Meanwhile, improving the efficiency of land use, the

simulation effect, and the achievability of the model iteration results was considered. Refer to the study of Chen et al. [18]. After repeatedly modifying the constraints and model parameters, the weights were set to $\varphi_1 = 0.47$, $\varphi_2 = 0.53$.

(2) Determination of constraints: The future land use simulation under the multi-scenario model was constrained by the laws of natural development and government planning expectations. Combined with the existing literature, this paper has limited the number of different land types in Zhengzhou in 2035 according to the "General Plan for Land Use in Zhengzhou (2006–2020)" and the trend of LUC from 2010 to 2020, and constraint conditions were constructed (Table 3).

### 2.3.3. PLUS-SA Model

Combining simulations, knowledge discovery, and decision-making, the patch-generating land use simulation (PLUS) model provides researchers and planning authorities with an important decision-making tool. The model integrates the rule-mining method based on the land expansion analysis strategy (LEAS) and the CA model based on multi-type random patch seeds (CARS) (Figure 4), which can be utilized to mine the driving factors of land expansion and predict the patch-level evolution of land use landscapes. Additional details regarding the PLUS model can be found at https://github.com/HPSCIL/Patch-generating_Land_Use_Simulation_Model, accessed on 12 June 2023.

To reasonably describe the dynamic process of the spatial distribution of LUC, the spatial autocorrelation of land use data must be comprehensively considered [52]. But, when PLUS uses RF to fit the relationship between land use and driving factors and then calculate the probability of an occurrence of each species, it does not consider the impact of possible spatial autocorrelation in land use data. This paper intends to introduce the spatial autocorrelation factor of different land types as the RF model's input variable to improve the accuracy of the PLUS model; we call it the PLUS model considering Spatial Autocorrelation (PLUS-SA). Specifically, in the LESA module of the PLUS model (Figure 4), we input the spatial autocorrelation factors of land use types into the RF model as driving factors to obtain the growth possibilities of different land use types, which will potentially help to improve the accuracy of model simulation. The spatial autocorrelation factor can be expressed as:

$$Autocov_p = \frac{\sum_{p \neq j} w_{pj} y_j}{\sum_{p \neq j} w_{pj}} \qquad (10)$$

where $Autocov_p$ represents the spatial autocorrelation variable of land type '*i*' on the cell $p$; *j* represents the neighborhood cell of cell $p$; $y_j$ represents the occurrence of land type '*i*' on cell *j* (it is assigned a value of 1 when it occurs, otherwise, it is 0); and $w_{pj}$ is the spatial weight coefficient between cell $p$ and cell *j*. When the distance between $p$ and *j* is less than the threshold distance $d$, $w_{pj} = 1/d$; otherwise, $w_{pj} = 0$.

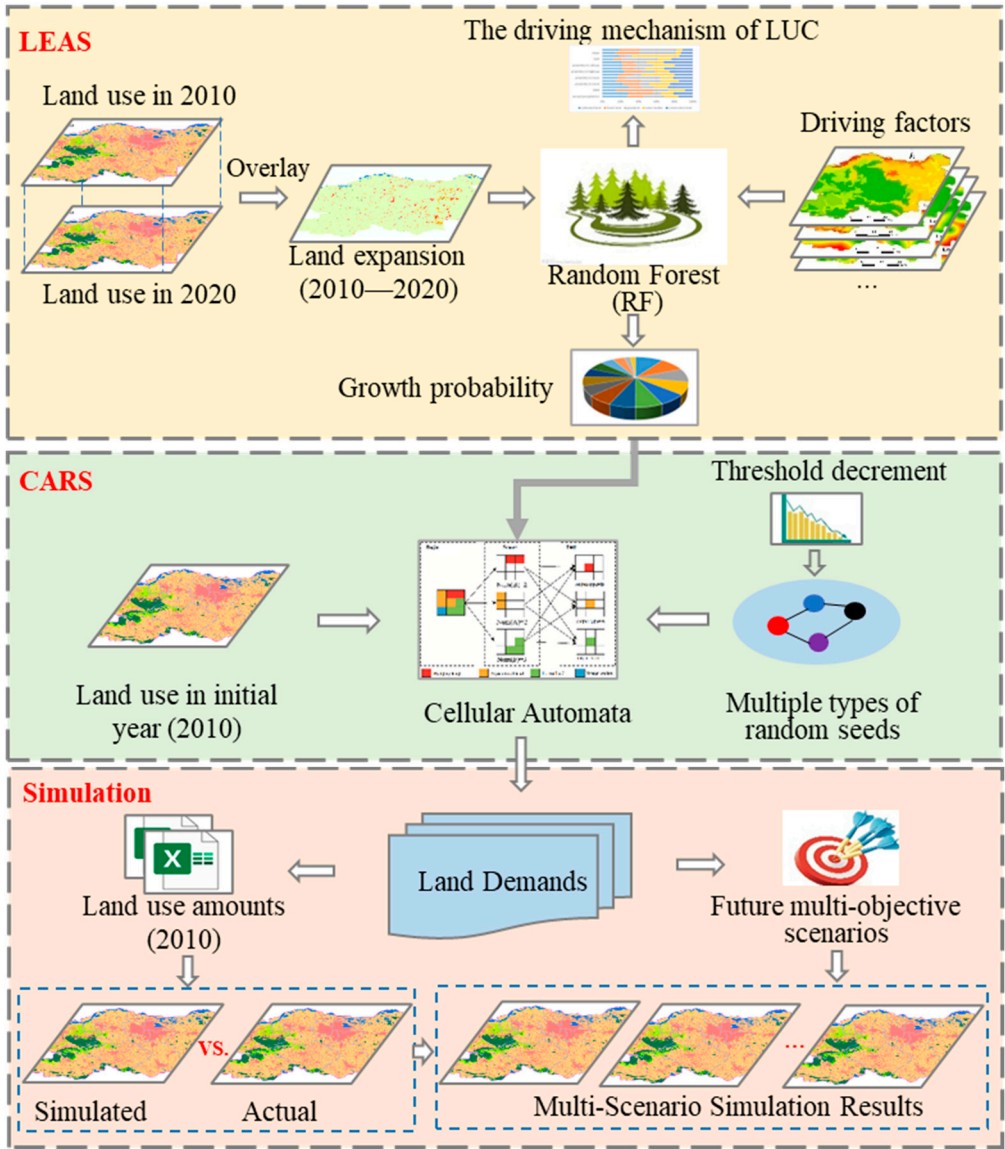

**Figure 4.** The PLUS model framework.

### 2.3.4. Model Validation Method

This study validated the model's accuracy by comparing actual 2020 land use data with simulated data from FLUS, PLUS, and PLUS-SA. Model accuracy verification tests were performed using the following two methods. First, the model was evaluated quantitatively by establishing a confusion matrix between the simulated land use results and the corresponding grid cells of the actual land use, and then calculating the simulated *Kappa* and overall accuracy. The Kappa coefficient is an index used for consistency testing and can also be used to measure the effect of classification, because, for classification problems, the so-called consistency is whether the model prediction results and the actual classification results are consistent. The calculation formulas are as follows:

$$Kappa = \frac{OA_o - OA_e}{1 - OA_e} \tag{11}$$

$$OA_O = (\sum_{k=1}^{n} OA_{kk})/N \tag{12}$$

where $OA_O$ is the overall accuracy, which represents the probability that the simulation results of each random sample are consistent with the real land use data, and its value is equal to the ratio of the number of correct pixels to the total number of pixels; $n$ is the number of land use types, which was 6 in this study; N is the total number of samples; $OA_{kk}$ indicates the quantity of correctly classified samples of land type k; and $OA_e$ represents the accidental consistency between the predicted and actual land use data. The closer the overall accuracy and *Kappa* are to 1, the better the simulation results and the higher the model accuracy.

Second, the consistency of the spatial location was verified by calculating the figure of merit (FOM) index. Compared with *Kappa*, the *FOM* can better describe the accuracy of land use simulations. The *FOM* index was utilized to verify the consistency between the simulated change and the real change. The range is 0 to 1, and the larger the value, the more the simulated variation and the real variation overlap. The calculation formula can be expressed as follows:

$$FOM = \frac{B}{A + B + C + D} \tag{13}$$

where $A$ is the real data change, but was simulated as a constant grid cell; $B$ is the number of grid cells where the real data change, the simulation results also change, and the changes are consistent; $C$ is the number of grid cells where the real data change and the simulation results also change, but the change is an incorrect category; and $D$ is the number of grid cells in which the real data do not change, but are simulated as changing.

## 3. Results

### 3.1. LUC between 2000 and 2020

Table 4 shows that the main land use type in Zhengzhou was cultivated land, followed by construction land, forestland, grassland, and water bodies. In 2020, the cultivated and construction land areas accounted for 56.32% and 26.95% of Zhengzhou's total area, respectively. The areas of forests, grassland, and water bodies were comparable, accounting for between 4% and 7.5% of Zhengzhou's total area. It is worth noting that the small amount of unused land in 2000 gradually disappeared over time. From the perspective of single land use dynamics, the *K* values of cultivated land, forestland, grassland, and unused land were all negative during the different monitoring periods (2000–2010, 2010–2020, and 2000–2020), while those of water bodies and construction land were all positive. This indicates that the areas of cultivated land, forestland, grassland, and unused land in Zhengzhou had continued to decrease, and the areas of water bodies and construction land had continued to increase; the construction land area, in particular, had increased the most (83.85% of the newly added construction land came from cultivated land) over the past 20 years. The forestland area decreased by 197.99 km$^2$ from 2000 to 2010; however, this may have been affected by the policy of returning farmland to forests, and the forest area decreased slightly from 2010 to 2020 (decreasing by 3.69 km$^2$). Judging from the spatial visualization results of the comprehensive land use dynamic degree (Figure 5), the high value of the comprehensive land use dynamic degree was mainly distributed around the main urban area, as well as the central and northern mountainous areas in Zhengzhou during the 2000 to 2010 period, which may mainly have been due to the utilization of land resources by human activities, such as urban expansion, deforestation, and wasteland cultivation. The implementation of the "Zhengzhou–Kaifeng Integration" and the second-phase expansion project of the Zhengzhou Xinzheng International Airport were the primary reasons for the high spatial distribution of the comprehensive land use dynamic degree in the eastern part of the main urban area of Zhengzhou and the Zhengzhou Airport Economy Zone between 2000 and 2010. Throughout the research period (2000–2020), urban expansion around the main urban area was evident in Zhengzhou.

**Table 4.** Land use structure and single land use dynamic degree in Zhengzhou.

| Land Use Type | | Cultivated Land | Forestland | Grassland | Water Bodies | Construction Land | Unused Land |
|---|---|---|---|---|---|---|---|
| 2000 | Area (km$^2$) | 5060.81 | 762.64 | 690.02 | 206.05 | 844.97 | 3.31 |
| | Percentage (%) | 66.87% | 10.08% | 9.12% | 2.72% | 11.17% | 0.04% |
| 2010 | Area (km$^2$) | 4629.13 | 564.65 | 398.48 | 281.92 | 1693.62 | 0.00 |
| | Percentage (%) | 61.17% | 7.46% | 5.27% | 3.73% | 22.38% | 0.00% |
| 2020 | Area (km$^2$) | 4262.44 | 560.96 | 392.27 | 312.90 | 2039.23 | 0.00 |
| | Percentage (%) | 56.32% | 7.41% | 5.18% | 4.13% | 26.95% | 0.00% |
| | 2000–2010 | −0.85% | −2.60% | −4.23% | 3.68% | 10.04% | −10.00% |
| *K* | 2010–2020 | −0.79% | −0.07% | −0.16% | 1.10% | 2.04% | 0.00% |
| | 2000–2020 | −0.79% | −1.32% | −2.16% | 2.59% | 7.07% | −5.00% |

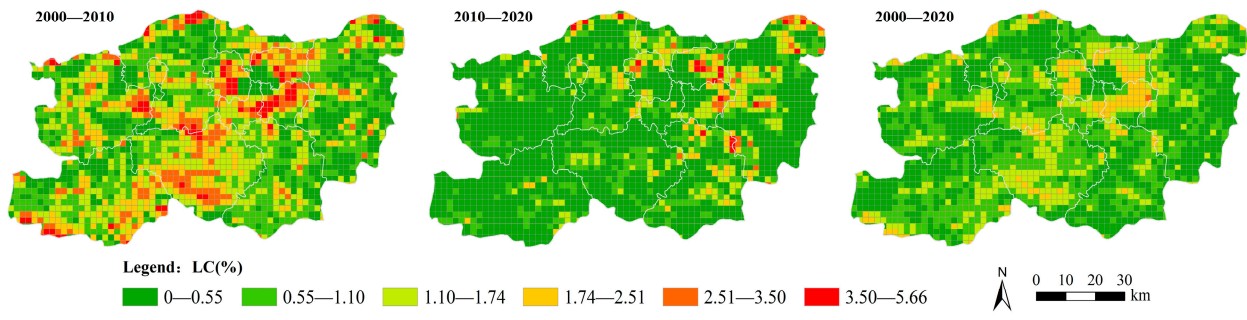

**Figure 5.** Comprehensive land use dynamic degree in Zhengzhou.

### 3.2. Simulation of the Demand for Land Use Structures in Multiple Scenarios

The land use quantity demand in 2035 was determined under four scenarios using the Markov and MOP (based on Lingo12.0 software and Markov Chain tool) models (Table 5). The cultivated land would decrease by approximately 554 km$^2$ under the ND and ED scenarios, and the construction land would continue to increase, with the largest increase compared with other scenarios. Meanwhile, compared with other scenarios, the area of water bodies would decrease by a maximum of 30.22 km$^2$ under the ED scenario, seriously threatening the regional ecological environment. Under the EP scenario, the reduction in cultivated land would be at its smallest (234.85 km$^2$) and the growth rate of construction land would be effectively limited. On the contrary, the areas of forestland, grassland, and water bodies with high ecosystem service value coefficients would increase significantly compared with those in 2020, with increases of 111.82 km$^2$, 6.55 km$^2$, and 62.46 km$^2$, respectively. Under the SD scenario, changes in different land types took into account the dual goals of economic development and ecological protection. Cultivated land would be consistent with the ND and ED scenarios on the premise of meeting the population demand in 2035, while the forestland, grassland, and water body areas would increase significantly compared with those under the two scenarios, but the increase would be slightly smaller than that under the EP scenarios. The increase in the construction land area with a higher economic value coefficient would be larger than that of the EP scenario, but it would also be controlled to a certain extent compared with the ND and ED scenarios. Judging from the results of Zhengzhou's economic and ecological benefits under different scenarios, the ND and ED scenarios targeted economic development, and the economic value would be about $1.09 \times 10^4$ billion CNY higher than that in 2020, but the ecological benefit would also decrease by 7.32 billion CNY. Under the EP scenario, with ecological protection as the primary goal, although the ecological benefit of Zhengzhou would increase by 6.9 billion CNY compared with that in 2020, economic development will be slow, only increasing by $0.08 \times 10^4$ billion CNY. In contrast with the above three scenarios, the economic and ecological benefits under the SD scenario will both increase compared with those in 2020.

Among them, the economic benefits would increase by $0.75 \times 10^4$ billion CNY and the ecological benefits would increase by 1.71 billion CNY.

**Table 5.** The land use demand (km$^2$) and economic/ecological benefits (CNY) under different scenarios.

| Type | 2020 Actual | 2035 Land Use Demand | | | |
|---|---|---|---|---|---|
| | | ND Scenario | ED Scenario | EP Scenario | SD Scenario |
| Cultivated land | 4262.44 | 3708.79 | 3708.97 | 4027.59 | 3708.97 |
| Forestland | 560.96 | 552.99 | 560.77 | 672.78 | 667.56 |
| Grassland | 392.27 | 377.96 | 398.82 | 398.82 | 378.96 |
| Water bodies | 312.90 | 311.06 | 282.68 | 375.36 | 363.55 |
| Construction land | 2039.23 | 2616.99 | 2616.54 | 2093.25 | 2439.76 |
| Economic benefits (billion) | $3.96 \times 10^4$ | $5.04 \times 10^4$ | $5.05 \times 10^4$ | $4.06 \times 10^4$ | $4.71 \times 10^4$ |
| Ecological benefits (billion) | 105.2 | 97.88 | 96.42 | 112.01 | 106.91 |

*3.3. The Accuracy Comparison of PLUS-SA, PLUS, and FLUS Models*

Based on the two phases (2010 and 2020) of land use and other geographic data, this study simulated the land use layout in 2020 using the FLUS, PLUS, and PLUS-SA models and calculated the *Kappa* and *FOM* in order to verify the accuracy of the three models. When simulating future land use patterns, 16 driving factors were selected by referring to the existing research results [19] and considering the accessibility, systematicness, and representativeness of the driving factors. Among them, the six natural driving factors include: DEM, slope, aspect, annual mean temperature, annual precipitation, and soil type. Further, the ten socioeconomic driving factors include population, GDP, proximity to a highway, proximity to a railway, proximity to a trunk road, proximity to a primary road, proximity to a secondary road, proximity to a tertiary road, proximity to a town, and proximity to open water. The parameters of the LEAS and CARS modules of the PLUS-SA have been tested on many occasions, and the regression tree number and sampling rate were finally set to 50 and 0.05, respectively, in the LEAS module. In the CARS module, the patch generation threshold, expansion coefficient, and percentage of seeds were set to 0.9, 0.1, and 0.0001, respectively. The $3 \times 3$ Moore neighborhood was adopted to quantify the neighborhood effects of the PLUS model. The land use transition matrix was set with reference to the actual transition matrix from 2010 to 2020, and neighborhood weights were established in reference to the study by Wang et al. [62]. Meanwhile, some policies restricted changes in all land types in designated areas, such as nature reserves, open waters, etc. This study set water bodies and national nature reserves as restricted transformation zones.

Figure 6 displays a contrast of the simulated land use patterns utilizing the FLUS (a1), PLUS (b1), and PLUS-SA (c1) models. Panels a2, b2, and c2 show the incorrect patches of the result for 2020 simulated using the three models compared with the actual land use pattern in 2020. Panels a3, b3, and c3 show the incorrect patches of a subregion. Table 6 displays the accuracy of the different models. We can see that the Kappa was above 0.85 for all three models, and the overall accuracy was over 90%. The accuracy of the PLUS model was higher than that of the FLUS model. Meanwhile, the Kappa, overall accuracy, and the FOM of the PLUS-SA model were higher than those of the PLUS and FLUS models. The PLUS-SA model had significantly fewer misclassified plaques than the other two models, according to the comparison of a3, b3, and c3 in Figure 6. Therefore, the simulation accuracy of the PLUS-SA model was better than that of the FLUS model and traditional PLUS model, and better simulation results concerning LUC based on the PLUS-SA model could be achieved.

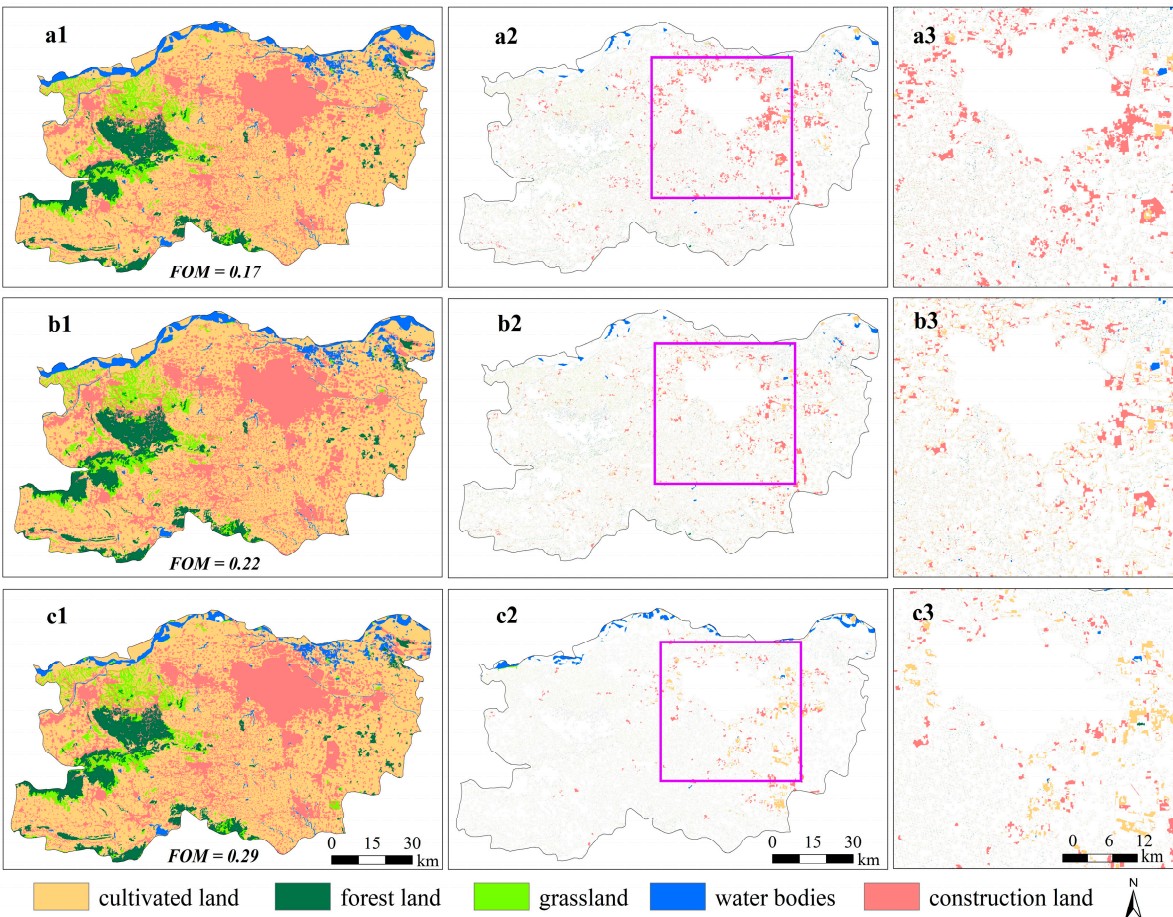

**Figure 6.** Comparison of the simulated land use patterns using the FLUS, PLUS, and PLUS-SA models. (Panels **a1**,**b1**,**c1** representative the simulated land use patterns in 2020 by utilizing the FLUS, PLUS, and PLUS-SA models, respectively. Panels **a2**,**b2**,**c2** show the incorrect patches of the result for 2020 simulated using the three models compared with the actual land use pattern in 2020. Panels **a3**,**b3**,**c3** show a subregion of the purple box in **a2**,**b2**,**c2**, respectively), to make it easier for readers to understand our manuscript.

**Table 6.** FLUS model, PLUS model, and PLUS-SA model accuracy results.

| Model Type | *Kappa* | **Overall Accuracy** | *FOM* |
|------------|---------|----------------------|-------|
| FLUS model | 0.85 | 0.90 | 0.17 |
| PLUS model | 0.87 | 0.92 | 0.22 |
| PLUS-SA model | 0.91 | 0.95 | 0.29 |

*3.4. Spatial Layout of Future Land Use for Different Scenarios*

The parameter settings for the Zhengzhou 2035 land use simulation under different scenarios were the same as those in Section 3.3, except for the transition matrix in the CARS module. The land use transfer constraints were different under different scenarios. Specifically, the transition matrix setting for the ND scenario was the same as that in Section 3.3. Under the ED scenario, construction land can be expanded to any other land type. Under the EP scenario, the government pays more attention to the protection of land types with high ecosystem service value coefficients, for instance, water bodies and forestland. Therefore, it was stipulated that land types with high ESV coefficients were prohibited from being converted into land types with lower ESV coefficients. Under the SD scenario, the expansion of urban construction land and the protection of water bodies and forestland with high ESV coefficients were taken into account. The transition matrix under

different scenarios is shown in Table 7, where 0 signifies that a conversion was prohibited and 1 signifies that a conversion was allowed.

**Table 7.** The land use transition possibility matrix for different scenarios.

| Scenario | ED | | | | | EP | | | | | SD | | | | |
|---|---|---|---|---|---|---|---|---|---|---|---|---|---|---|---|
| | CL | FL | GL | WB | CTL | CL | FL | GL | WB | CTL | CL | FL | GL | WB | CTL |
| CL | 1 | 1 | 1 | 0 | 1 | 1 | 1 | 1 | 0 | 1 | 1 | 1 | 1 | 0 | 1 |
| FL | 1 | 1 | 0 | 0 | 1 | 0 | 1 | 0 | 0 | 0 | 0 | 1 | 1 | 0 | 0 |
| GL | 1 | 1 | 1 | 1 | 1 | 0 | 1 | 1 | 0 | 0 | 0 | 1 | 1 | 1 | 1 |
| WB | 1 | 0 | 1 | 1 | 0 | 0 | 0 | 0 | 1 | 0 | 1 | 0 | 0 | 1 | 0 |
| CTL | 0 | 0 | 0 | 0 | 1 | 1 | 1 | 1 | 0 | 1 | 0 | 0 | 1 | 0 | 1 |

Abbreviations: cultivated land (CL); forestland (FL); grassland (GL); water bodies (WB); construction land (CTL).

Based on the PLUS-SA model, the land use spatial distribution of Zhengzhou was simulated under four scenarios in 2035, and a land use expansion map from 2020 to 2035 was produced (Figure 7). The results showed that construction land expansion would be the most significant under the ND and ED scenarios. The newly added construction land would primarily be located around the main urban area in Zhengzhou, the Zhengzhou Airport Economy Zone, and Zhongmu County to the east of Zhengzhou City. As the Zhengdong New District, Zhengzhou International Cultural and Creative Industry Park, Zhengzhou Economic Development Zone, Zhengzhou Airport Zone, and so on have all mainly been distributed in most of the towns in the west (southwest and northwest) of Zhongmu, most of Zhongmu County has, in fact, been gradually forming a "same city" along with the main city of Zhengzhou. Under the EP scenario, the expansion of construction land would be the slowest; in contrast, the forestland to the southwest of Zhengzhou, which is unsuitable for human habitation and less disturbed by human activities because of its high altitude and complicated topography, would increase significantly. Under the SP scenario, the expansion of construction land and forestland would be relatively obvious, and the spatial location of the expansion would be highly consistent with the above-mentioned ED and EP scenarios.

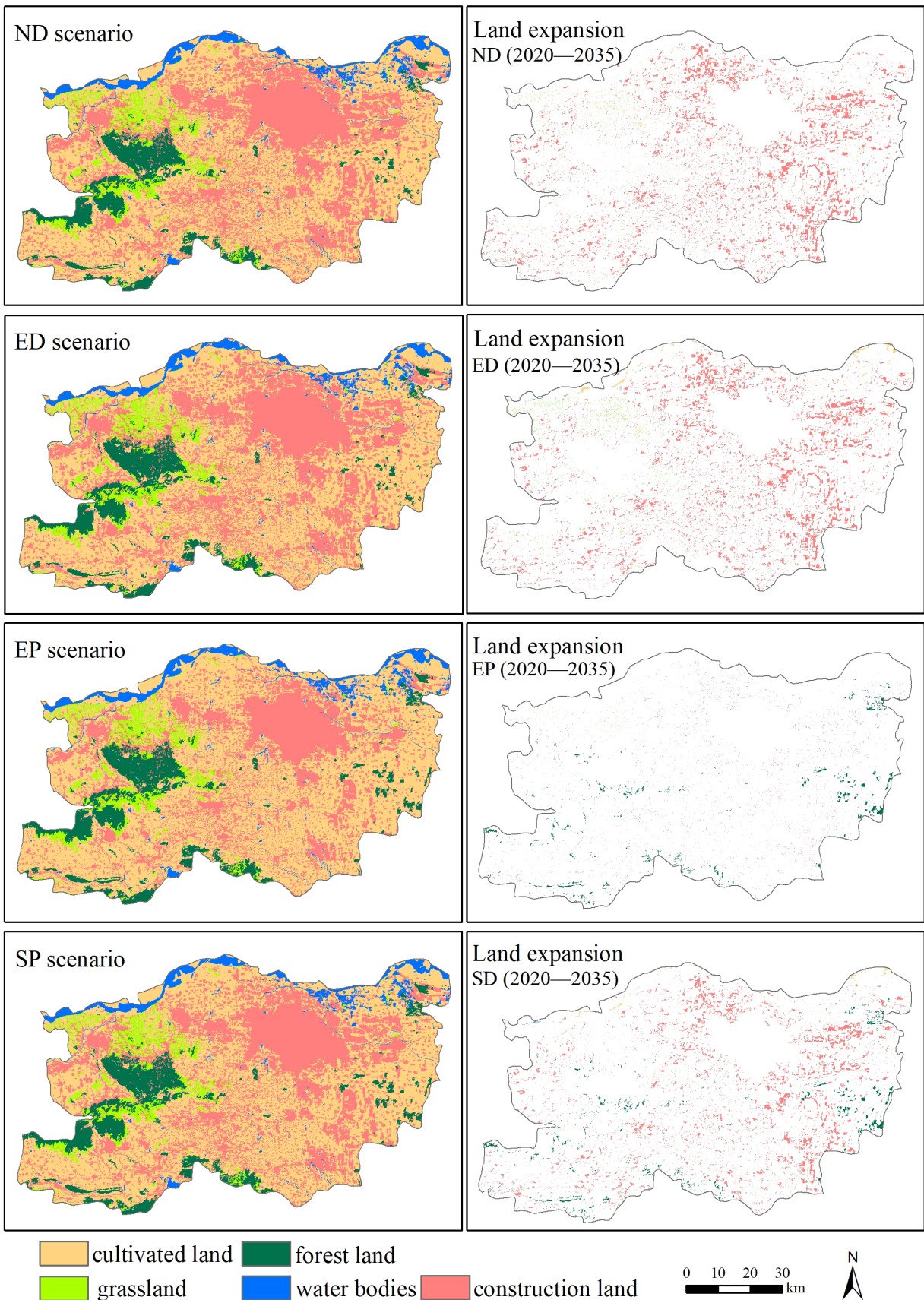

**Figure 7.** Land use spatial distribution and land use expansion (2020–2035) under different scenarios.

## 4. Discussion

### 4.1. Urban Expansion-Exacerbated LUC

Urban expansion is a necessary process for the development of cities of different scales in China and even in the world [63]. The "man–land" contradiction and "space conflict" caused by urban expansion are bound to attract the attention of policymakers and researchers [64,65]. As a unipolar core city in Henan, Zhengzhou is also a key city for population agglomeration. With its continuous population growth, the land around Zhengzhou City has been converted into construction land, thus increasing the speed of urban expansion. Relevant studies have shown that the built-up central urban area of Zhengzhou is expanding at a scale of nearly 70 km$^2$ every year [66], which means that Zhengzhou's fundamental rapid population growth and urban expansion will not change. Urban expansion must come at the expense of encroaching on other land resources, which, in turn, drives regional land use changes [67]. Our research shows that 83.85% of the land used for urban expansion in Zhengzhou from 2000 to 2020 came from cultivated land, directly threatening food security and hindering sustainable agricultural development. Meanwhile, forestland and grassland areas with higher ecosystem service value coefficients decreased by 201.68 km$^2$ and 297.75 km$^2$, respectively, from 2000 to 2020, which also brought huge challenges to the regional ecological environment. It is concluded here that Zhengzhou is currently facing the following problems: (1) the construction land area is growing rapidly, and there is a conflict between protecting cultivated land and ensuring development; (2) the cultivated land area per capita is continuing to decline, and the reserve resources for cultivated land are insufficient; (3) the contradiction between land exploitation and utilization, ecosystem construction, and environmental conservation has intensified, and the task of coordinating protection and development has become more arduous. Therefore, policymakers should coordinate the relationship between cultivated land protection, ecosystem services, and urban development. They should scientifically delineate spatial control boundaries, such as permanent basic farmland protection red lines, ecological protection red lines, and urban growth boundaries [68,69]. The ultimate target is to optimize the spatial distribution of land use and achieve the goal of intensive and efficient land resource utilization and sustainable development.

### 4.2. Coupling Model Contributing to the Sustainable Development of Land Resources

Clarifying the processes and laws of LUC and identifying the key factors that drive LUC are of great significance, particularly with regard to adjusting land use policies and realizing efficient, green, and intensive uses of land resources [70,71]. In the LESA module of the PLUS model, firstly, based on the 2000 and 2020 land use data, the expansion grids of various land types were extracted and sampled, and then random forest was utilized to mine various land use expansion and driving factors one by one. Finally, the contributions of driving factors to various land use expansions were captured during the monitoring period between 2000 and 2020 based on Origin software (Figure 8). It was found that annual precipitation, annual mean temperature, and DEM rank were the three most significant factors when it came to the contribution rate to cultivated land expansion, followed by the proximity to roads, which indicated that natural factors greatly influenced cultivated land expansion, and the newly increased cultivated land was more distributed in areas with moderate temperatures, abundant precipitation, and convenient transportation. With regard to forestland, DEM and population were the key driving factors behind forestland expansion, and their contribution rates were 0.2418 and 0.1329, respectively, indicating that newly increased forestland was mainly distributed in mountainous areas with higher altitudes and lower impact from human activities. GDP, proximity to a town, and population contributed the most to grassland changes, and their values all exceeded 0.1. The newly increased grasslands were most likely to be distributed in areas that were far from towns with higher GDP and were less affected by human activities. The primary factors driving the expansion of water bodies were proximity to open water, soil type, and DEM. Construction land expansion was principally distributed in regions near towns with

high GDP. Therefore, the driving factors behind the expansion of different land types were different, and this should be taken into account by policymakers when formulating relevant land use policies [72].

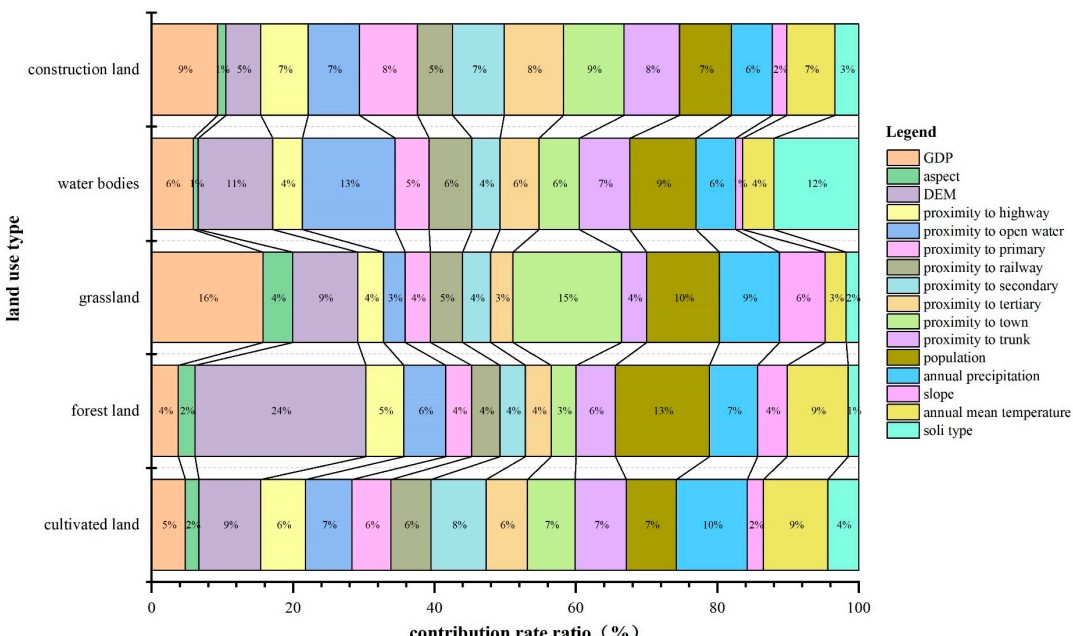

**Figure 8.** The contribution of driving factors to land use expansion from 2000 to 2020.

The land use structure demands (Table 5) that assisted in enhancing regional sustainable development were acquired and were significant in helping policymakers to determine future management directions and rational land use policies. For instance, the optimal land use structures for economic development and ecological protection were calculated under the ED and EP scenarios, respectively. Decision-makers can formulate different land use planning policies according to different target needs. However, these two development models only considered the land use structure needs of single-objective planning, which led to the pursuit of one land benefit while ignoring other land benefits. It was found here that, under the ED scenario, the economic benefits will increase the most compared with those in 2020 (1.09 × 10⁴ billion CNY), but the ecological benefits will decrease the most compared with those in 2020 (7.32 billion CNY). Similarly, under the EP scenario, although the ecological benefit exhibited the largest increase (6.81 billion CNY), economic development would be slow, and the economic benefit would only increase by 0.08 × 10⁴ billion CNY compared with that in 2020. Under the ED scenario, when decision-makers pursue economic benefits, the reductions in cultivated land, forestland, and grassland areas brought about by urban expansion would indirectly bring serious challenges to the ecological environment, while the EP scenario is not conducive to economic development. These are clearly not in line with the current requirements of high-quality and sustainable urban development. The SD scenario takes into account the dual goals of economic development and ecological protection. Compared with those in 2020, the economic and ecological benefits will increase by 0.75 × 10⁴ billion CNY and 1.71 billion CNY, respectively, in 2035. Therefore, coupling MOP and PLUS-SA can lead to seeking a sustainable future land use spatial layout scenario. However, there are many factors that affect the performance of the coupled MOP and PLUS-SA model, which need to be taken into account during the research process. For example, the government's future land use planning policy will affect the performance of the MOP model by influencing the land use structure. The setting of computer hardware and LEAS and CARS module parameters directly affects the simulation speed and accuracy of the PLUS-SA model.

*4.3. Limitations and Future Directions*

Scientifically understanding and coordinating a "human–land relationship" is an essential condition for achieving sustainable development. While land use is the core issue of a human–land relationship, LUC is also a vital way to study the human–land relationship [73,74]. This study considered the effects of human activities on LUC, which were reflected in the simulation process of LUC in the form of driving factors. However, LUC also influences human activity and cognition, which is rarely seen in modeling work [75]. Future land use simulations are expected to start from the perspective of the "human–land" interaction. By exploring the interaction between LUC and human activities, the establishment of a scientific concept of land use will be promoted, and human behavior will be adjusted and restrained. Moreover, most existing land use simulation studies regard regions as isolated individual units, ignoring the teleconnection effect of land use between different regions [76]; that is, owing to the economic division of labor and location advantages, economic activities and consumer demand within a region are often no longer supported by local land use, but are instead transferred to other regions through inter-regional trade activities, thus causing changes in land use within other regions [77]. Therefore, using economic quantitative models, such as the multi-regional input–output (MRIO) model, to quantify the teleconnection effect of land use between regions and then coupling a spatial model to simulate the future spatial distribution of land use will be a new direction for land use simulation research.

## 5. Conclusions

We proposed the PLUS-SA model by introducing the spatial autocorrelation factor as the input variable for the RF model in the LEAS module of the PLUS model and verified the simulation accuracy of the PLUS-SA model. The LESA module was then used to explore the driving mechanism of LUC in Zhengzhou from 2010 to 2020. Finally, the land use quantitative demand and spatial distribution were predicted under the ND, ED, EP, and SD scenarios in Zhengzhou City based on the coupled MOP and the PLUS-SA model. The following important conclusions were reached. (1) With accelerations in population growth and urbanization in Zhengzhou, urban expansion is still proceeding rapidly. Urban expansion has encroached on a great deal of cultivated land (83.85% of the land for urban expansion from 2000–2020 was cultivated land) and other ecological land in Zhengzhou, and the task of coordinating ecological protection and economic development was arduous. (2) Natural factors (topography, temperature, and precipitation) greatly influenced cultivated land expansion. Socioeconomic factors (population, GDP, and proximity to a town) had a high contribution rate to grassland and construction land expansion. However, forestland expansion was the result of a combination of natural factors (DEM) and socioeconomic factors (population). (3) The *Kappa* coefficient, overall accuracy, and *FOM* of the PLUS-SA model were 0.91, 0.95, and 0.29, respectively, and higher than those of the traditional PLUS and FLUS models, indicating that the PLUS-SA model had a higher degree of simulation accuracy. (4) Land use structures and spatial distributions under four scenarios in 2035 and their corresponding economic and ecological benefits can be forecasted by coupling the MOP and PLUS-SA models. Such forecasts will be important for policymakers when formulating sustainable land resource development policies for the study area. Notably, the land use structure and spatial distribution under the SD scenario can be used as criteria for inspecting whether Zhengzhou City has developed along a sustainable pathway.

The PLUS-SA model we proposed could be applied to the study of geospatially intelligent simulation, and the findings of this study can provide a reference for the sustainable development of land resources in Zhengzhou. However, in the process of land use simulation, not only the influence of human activities on LUC, but also the influence of LUC on human activities and cognition should be considered, which is rarely seen in modeling work. Additionally, the existing land use simulation studies regard the region as an isolated individual, and few consider the influence of an external region on the study

area. The above two points can be used as new directions for future land use simulation research. In spite of this, our research framework can provide a basis for decision-makers to formulate sustainable land use development policies to realize high-quality and sustainable urban development.

**Author Contributions:** Conceptualization, P.G. and F.Z.; methodology, P.G.; software, F.Q.; validation, P.G., F.Z. and C.M.; formal analysis, F.Q.; investigation, C.M.; resources, C.M.; data curation, P.G.; writing—original draft preparation, P.G.; writing—review and editing, F.Z.; visualization, F.Q.; supervision, H.W.; project administration, H.W.; funding acquisition, H.W. All authors have read and agreed to the published version of the manuscript.

**Funding:** This research was funded by the National Major Project of High-Resolution Earth Observation System, Grant/Award Number: 80-Y50G19-9001-22/23; University Young Key Teacher Training Plan of Henan Province, Grant/Award Number: 2020GGJS028; Natural Science Foundation of Henan, Grant/Award Number: 202300410096; Key Scientific Research Project Plans of Higher Education Institutions of Henan, Grant/Award Number: 21A170008; Technology Development Plan Project of Kaifeng, Grant/Award Number: 2003009.

**Data Availability Statement:** Data sharing not applicable.

**Acknowledgments:** The authors would like to thank the editor and reviewers of the manuscript for their thoughtful and helpful comments.

**Conflicts of Interest:** The authors declare no conflict of interest.

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
