# Peer review of "Coupled MOP and PLUS-SA Model Research on Land Use Scenario Simulations in Zhengzhou Metropolitan Area, Central China"

_remotesensing, doi:10.3390/rs15153762_

Round 1
Reviewer 1 Report
The paper is of good quality. It discussed the improvement of the land use prediction model PLUS. There are a few comments about the original manuscript.
1. It is suggested to decrease the numbers of figures and tables in the paper for the sake of the focus and lack of redundancy. Fig.5, Fig.7, Fig.10 could be cancelled. Tabel.4 and Fig.6 could be selected either one .
2. To give the detailed explanation of the symbols in the figures. For example, in Fig.8, under the title of the figure(Line 439), the meaning of a1,b1,c1…… a3,b3,c3 may be explained(Line 426-429). The same solution is for Fig.7.
3. There are some conflicts in table 7 of the land use transition matrix in different scenarios. Under EP, CTL can be transformed into CL, FL, GL, it seems impossible. The table title could be changed as “ the land use transition possibility matrix….” . In fact it is not the transfer probability matrix for land use.
4. In Fig.4, how is the spatial correlation coefficient embedded into the PLUS model? Please explain it clearly. In Fig.4, is it proper to use “Land change rules” in place of “land demand” ?
5. As for the economic benefit coefficient of each land class in line 241-245, it is necessary to give the reasonable references.
6. At line 266& 267, the weights were set to ?1=0.47, ?2 = 0.53. Are the weights calculated or self-determined? Please give clear assertion.
The English is fluent and clear. I would like to know more clearly about model PLUS-SA from line 276 to 301.
Author Response
The authors would like to thank Chief Editor, Associate Editor and anonymous Reviewers for their constructive comments and suggestions to improve the quality of the paper. Those comments are all valuable and very helpful for revising and improving our paper. We have studied comments carefully and have made correction which we hope meet with approval. The main corrections in the paper and the responds to the reviewer’s comments are as flowing:
Response to Reviewer #1:
The paper is of good quality. It discussed the improvement of the land use prediction model PLUS. There are a few comments about the original manuscript.
Comment #1: It is suggested to decrease the numbers of figures and tables in the paper for the sake of the focus and lack of redundancy. Fig.5, Fig.7, Fig.10 could be cancelled. Tabel.4 and Fig.6 could be selected either one.
Response: Thanks for such a constructive comment. According to your comment, we have removed Figure 5 and Figure 7. However, Figure 10 reflects the contribution of each driving factor to the expansion of different land use types. If Figure 10 is deleted, the content of lines 530-545 will lack analysis basis, so we do not recommend deleting it. Again, Table 4 quantifies the land use structure and single land use dynamic degree, and Figure 6 is the spatial visualization of the comprehensive land use dynamic degree in Zhengzhou. There is no redundancy between them, and removing either would be a lack of basis for the results of the article.
Comment #2: To give the detailed explanation of the symbols in the figures. For example, in Fig.8, under the title of the figure (Line 439), the meaning of a1, b1, c1…… a3, b3, c3 may be explained (Line 426-429). The same solution is for Fig.7.
Response: Thank you for your careful inspection. We have described and explained the meanings of a1, b1, c1; a2, b2, c2 and a3, b3, c3, respectively in lines 440-443 of the "Revised Manuscript". We have removed Figure 7 according to your comment #1.
Comment #3: There are some conflicts in table 7 of the land use transition matrix in different scenarios. Under EP, CTL can be transformed into CL, FL, GL, it seems impossible. The table title could be changed as “the land use transition possibility matrix….”. In fact, it is not the transfer probability matrix for land use.
Response: Thank you for your careful inspection. As you said, it seems impossible that CTL transformed into CL, FL, GL Under EP scenario, and we all agree that it is very appropriate that you suggest changing the title of Table 7 to "the land use transition possibility matrix in different scenarios". Therefore, we have made changes in accordance with your comments, please refer to Lines 470 for details of the "Revised Manuscript".
Comment #4: In Fig.4, how is the spatial correlation coefficient embedded into the PLUS model? Please explain it clearly. In Fig.4, is it proper to use “Land change rules” in place of “land demand”?
Response: Figure 4 shows the framework of the PLUS model. In the LESA module of the PLUS model, we input the spatial autocorrelation factors of land use types into the RF model as driving factors, and then obtain the growth possibilities of different land use types in this study. This helps to improve the accuracy of model simulation. We have already elaborated on line 314-317 of the "Revised Manuscript". Besides, I personally think it's feasible to use "Land change rules" because it's a bigger concept than "Land demand". However, land use simulation is divided into two parts: quantitative simulation and spatial simulation. This study mainly simulates the spatial layout of land use under the multi-objective planning scenario in the study area driven by the quantitative demand of land use. After reviewing the literature and consulting with model developers, we came to the conclusion that it is best to use the more specific concept of "land quantity demand driven".
Comment #5: As for the economic benefit coefficient of each land class in line 241-245, it is necessary to give the reasonable references.
Response: Thanks for such a constructive comment. The calculation method of economic benefit coefficient is indeed based on the corresponding references. It is necessary to cite the references. Thus, we have added the references used in calculating the economic benefit coefficient in line 264 of the "Revised Manuscript".
Comment #6: At line 266& 267, the weights were set to ?1=0.47, ?2 = 0.53. Are the weights calculated or self-determined? Please give clear assertion.
Response: The future development orientation of Zhengzhou City and the goal of simultaneously improving economic and ecological benefits were taken into account. Meanwhile, improving the efficiency of land use, the simulation effect, and the achievability of the model iteration results were considered. Refer to the study of Chen et al, and after repeatedly modifying the constraints and model parameters, the weights were set to ?1=0.47, ?2 = 0.53. Therefore, the setting of weights was determined based on the relevant literature and combined with the development goals of Zhengzhou to modify the constraints and model parameters. We have added references to line 287 of the "Revised Manuscript".
Comments on the Quality of English Language
- The English is fluent and clear. I would like to know more clearly about model PLUS-SA from line 276 to 301.
Response: Thank you for your English recognition of our manuscript, and our manuscript has been revised by a professional polishing agency. You would like to know more clearly about model PLUS-SA. In fact, the PLUS model is a classic model developed by Chinese scholar Liang, which has been widely used by domestic and foreign scholars in the study of intelligent simulation. Due to space constraints, the manuscript only briefly introduces the framework of the PLUS model. Additional details regarding the PLUS model can be found at https://github.com/HPSCIL/Patch-generating_Land_Use_Simulation_Model. The PLUS-SA model of this study considers the spatial autocorrelation effect between land use on the basis of the PLUS model, as detailed in lines 307-324 of the manuscript. Hopefully this will help you understand the PLUS-SA model more clearly.
Thanks again to the reviewers for their valuable suggestions for this study, and I wish you good health and success in your work.
Yours sincerely, all the authors.
May 26, 2023

Reviewer 2 Report
In this article, the PLUS-SA model was proposed to predict land use quantitative demand and spatial distribution in Zhengzhou City from 2010 to 2020.
The Research paper provides respectable findings and is well-written, but before it is accepted, it needs to be strengthened in the following ways:
In the abstract section, the author should include the qualitative results.
The introduction could be expanded, and more related research sources should be cited. The author may use the following sources: DOI: https://doi.org/10.3390/ijerph20054286, DOI: 10.3390/ijerph20031804, DOI: 10.1016/B978-0-443-15847-6.00010-0 and DOI: 10.13140/RG.2.2.10705.38246.
The author should include the True/False Color Composite Image in the methodology section.
The author should discuss the socioeconomic status of the study area.
The author needs to describe the Kappa Coefficient used for this study.
The author needs to write the full form of FLUS.
The author should describe the characteristics of the Satellite data used.
The author should mention the factors that affect the performance of the Coupled MOP and PLUS-SA model.
The author should define all abbreviations before using them even if they are well known.
The author should mention the name of the software/tools used for data analysis.
The author needs to include logical arguments for the findings, limitations, and directions for further research in the conclusion section.
Minor editing of the English language required
There is a typo error in many places, the author should correct them.
Author Response
The authors would like to thank Chief Editor, Associate Editor and anonymous Reviewers for their constructive comments and suggestions to improve the quality of the paper. Those comments are all valuable and very helpful for revising and improving our paper. We have studied comments carefully and have made correction which we hope meet with approval. The main corrections in the paper and the responds to the reviewer’s comments are as flowing:
Response to Reviewer #2:
In this article, the PLUS-SA model was proposed to predict land use quantitative demand and spatial distribution in Zhengzhou City from 2010 to 2020.
The Research paper provides respectable findings and is well-written, but before it is accepted, it needs to be strengthened in the following ways:
Comment #1: In the abstract section, the author should include the qualitative results.
Response: Thanks for such a constructive comment. We have accepted your comment and two qualitative conclusions were added at the end of the “Abstract”. Please refer to Lines 37-40 for details of the "Revised Manuscript ".
Comment #2: The introduction could be expanded, and more related research sources should be cited. The author may use the following sources: DOI: https://doi.org/10.3390/ijerph20054286, DOI: 10.3390/ijerph20031804, DOI: 10.1016/B978-0-443-15847-6.00010-0 and DOI: 10.13140/RG.2.2.10705.38246.
Response: Thank you for your constructive comment. The articles and books you recommended fit perfectly with my research topic, and it is necessary to quote them in the introduction to enrich the content of this part. We have cited these literatures and works in the "1. Introduction" part of the "Revised Manuscript".
Comment #3: The author should include the True/False Color Composite Image in the methodology section.
Response: Thanks for such a constructive comment. However, the remote sensing satellites have not yet been used in part of our methodology. Our research mainly uses data products produced based on remote sensing satellites, which have been described in the section "2.2. Data sources". Thus, should we include the True/False Color Composite Image in the methodology section? Maybe we didn't fully understand your intentions, and we would be pleased to have your further advice.
Comment #4: The author should discuss the socioeconomic status of the study area.
Response: Thanks for such a constructive comment. We have added a description of the social and economic situation of Zhengzhou in the "2.1 Study area" part of the manuscript. Please refer to Lines 148-153 for details of the "Revised Manuscript ".
Comment #5: The author needs to describe the Kappa Coefficient used for this study.
Response: Thank you for your constructive comment. In the research of land use simulation, Kappa coefficient is usually used to measure the consistency of model simulation results and actual classification results, and is a very common index. According to your suggestion, we have elaborated the Kappa coefficient in lines 331-334 of the "Revised Manuscript ".
Comment #6: The author needs to write the full form of FLUS.
Response: Thank you for your careful inspection. We have added the full form of FLUS to the “Abstract” where it first appeared. Please refer to Lines 30 for details of the "Revised Manuscript ".
Comment #7: The author should describe the characteristics of the Satellite data used.
Response: Thanks for such a constructive comment. The land use data set, which was the mainly data in this study, was mainly based on Landsat satellite remote sensing data and constructed by human–machine interactive visual interpretation. Thus, we have described the characteristics of the Landsat satellite in the "2.2 Data sources " part of the manuscript. Please refer to Lines 168-178 for details of the "Revised Manuscript ".
Comment #8: The author should mention the factors that affect the performance of the Coupled MOP and PLUS-SA model.
Response: Thanks for such a constructive comment. The government's future land use planning policy, parameter setting and computer hardware may affect the performance of the Coupled MOP and PLUS-SA model. We have described this in the "4.2 Coupling model contributing to the sustainable development of land resources" section of the manuscript. Please refer to Lines 572-577 for details of the "Revised Manuscript ".
Comment #9: The author should define all abbreviations before using them even if they are well known.
Response: Thank you for your constructive comment. We have checked the full text to make sure that all abbreviations have full form when they first appear.
Comment #10: The author should mention the name of the software/tools used for data analysis.
Response: Thank you for your suggestion. We have clarified the software or tools used for the research results. Please refer to Line236-237, Line391, Line530, etc. for details of the "Revised Manuscript ".
Comment #11: The author needs to include logical arguments for the findings, limitations, and directions for further research in the conclusion section.
Response: Thanks for such a constructive comment. We have added logical arguments for findings, limitations, and directions for further research in the conclusion section. Please refer to Lines 624-634 for details of the "Revised Manuscript ".
Comments on the Quality of English Language
- Minor editing of the English language required
Response: Thank you for your comment. Our manuscript has been revised by a professional polishing agency. If there are any other grammatical errors, please feel free to let us know.
- There is a typo error in many places, the author should correct them.
Response: We have re-examined the manuscript and corrected some errors.
Thanks again to the reviewers for their valuable suggestions for this study, and I wish you good health and success in your work.
Yours sincerely, all the authors.
July 26, 2023
